# Dietary Behaviors, Sugar Intake, and Public Awareness of Nutritional Labeling Among Young Adults: Implications for Oral and Systemic Health

**DOI:** 10.3390/nu18010091

**Published:** 2025-12-27

**Authors:** Catalina Iulia Saveanu, Paula Ilie, Daniela Anistoroaei, Livia Ionela Bobu, Alexandra Ecaterina Saveanu, Octavian Boronia, Loredana Golovcencu

**Affiliations:** 1Department of Surgical, Faculty of Dental Medicine, Grigore T. Popa University of Medicine and Pharmacy, 700115 Iasi, Romania; catalina.saveanu@umfiasi.ro (C.I.S.); iliepaula21@gmail.com (P.I.); livia.bobu@umfiasi.ro (L.I.B.); loredana.golovcencu@umfiasi.ro (L.G.); 2Faculty of Dental Medicine, Grigore T. Popa University of Medicine and Pharmacy, 700115 Iasi, Romania; saveanu.alexandra-ecaterina@d.umfiasi.ro (A.E.S.); oboronia@gmail.com (O.B.)

**Keywords:** nutritional literacy, simple carbohydrates, food labeling, preventive nutrition, oral health, young adults, health behaviors

## Abstract

**Background/Objectives**: Within public health and preventive nutrition, food labeling plays a critical role in supporting healthier dietary behaviors. This study aimed to evaluate the behaviors, perceptions, and nutritional literacy of young adults from Iași, Romania, regarding simple carbohydrates (SCHO) consumption and food label-reading habits. **Materials and Methods**: A cross-sectional survey was conducted between May–June 2023 using 20-item Likert-scale questionnaire completed by 150 participants aged 18–30 years. Statistical analysis included descriptive metrics, Chi-square tests, and Pearson’s correlation, with significance set at *p* ≤ 0.05. **Results**: The cohort consisted of 72% females (N = 108) and 28% males (N = 42), with 42.7% (N = 64) holding university degrees. Although 22% (N = 33) considered SCHO consumption highly important, only 13.3% (N = 20) frequently read nutrition labels (*p* ≤ 0.05). Dietary patterns showed that 27.3% primarily consumed sweets, while others combined sweets with carbonated beverages, dairy products, or whole grains; overall, 44% (N = 66) reported frequent sweet consumption. Label reading was highest for sweets (40.7%), lower for dairy products (19.3%) and soft drinks (9.3%). Additionally, 30.7% (N = 46) checked only expiration dates, whereas just 11.3% (N = 17) reviewed nutritional content. Trust in label accuracy was low: 48% (N = 72) expressed neutrality and 14% (N = 21) disagreed. Although 77.3% (N = 116) recognized the link between sugar intake and dental caries, only 23.3% (N = 35) felt well informed about oral health risks. Taste dominated food selection (68.7%), while nutritional value was cited by 16.7% (N = 25). **Conclusions**: Young adults from Iași demonstrated notable gaps in nutritional literacy and suboptimal dietary behaviors, emphasizing the need for structured educational strategies to improve preventive practices relevant to systemic and oral health.

## 1. Introduction

In recent years, growing attention has been directed towards the health consequences of excessive sugar intake, particularly its contribution to chronic metabolic disorders and oral diseases. Simple carbohydrates (SCHO), including monosaccharides (glucose, fructose, galactose) and disaccharides (sucrose, lactose, maltose), are widely present in modern diets. These sugars occur intrinsically in natural foods and extrinsically as added sugars in sweetened beverages, ultra-processed snacks, and baked goods [1,2].

Although carbohydrates play an essential role in sustaining physiological functions by providing energy to the brain, muscles, and nervous system, excessive consumption of refined SCHO has been associated with increased glycemic load, insulin resistance, and abdominal obesity [3,4]. Young adults aged 18–30 years are particularly vulnerable to these risks due to increased autonomy in food choices, exposure to targeted marketing strategies, and insufficient nutritional literacy.

From a public health perspective, the World Health Organization (WHO) recommends limiting free sugar intake to less than 10% of total daily energy intake, with a conditional recommendation of below 5% to further reduce the risk of metabolic diseases and dental caries [5,6,7,8,9,10]. Despite these guidelines, adherence remains low, and sugar-dense diets continue to be widespread across populations [11].

In the context of oral health, frequent intake of simple carbohydrates (SCHO) promotes the growth of acidogenic bacteria, such as *Streptococcus mutans*, leading to enamel demineralization and the development of dental caries [12,13,14,15]. These harmful effects are further exacerbated by inadequate oral hygiene practices and limited awareness of “hidden sugars” in processed foods.

While most food labeling systems require the disclosure of total carbohydrate and sugar content, they often fail to distinguish between naturally occurring and added sugars. Moreover, relevant indicators such as glycemic index and glycemic load are rarely reported, which limits the usefulness of food labels in managing metabolic and dental health risks [16]. Nutritional literacy—defined as the ability to understand and apply nutrition information—is essential for making healthier dietary choices. However, previous studies indicate that many young adults misinterpret food labels or underestimate the health implications of added sugars, particularly when products are marketed using claims such as “no added sugar” or “natural” [11,12].

Understanding how young individuals interpret and respond to food labels is crucial for designing effective nutritional education strategies and preventive healthcare programs. Therefore, the primary objective of this study was to evaluate the level of knowledge and behavioral practices related to simple carbohydrate (SCHO) consumption and food label interpretation among young adults aged 18–30 years in Iași, Romania.

While previous studies have examined sugar consumption patterns, nutrition label use, or oral health outcomes independently, fewer investigations have addressed these dimensions in an integrated manner within the same population. Moreover, existing research has largely focused on either general dietary behaviors or metabolic health outcomes, with limited attention to the intersection between nutritional literacy, food label interpretation, and oral health-related preventive behaviors in young adults.

The present study extends the current body of literature by simultaneously assessing knowledge, attitudes, and behaviors related to simple carbohydrate consumption and food labeling awareness, while explicitly considering their implications for oral health. By focusing on young adults aged 18–30 years, this research provides context-specific insights relevant to preventive dentistry and public health nutrition, thereby contributing to the development of targeted, personalized educational and preventive strategies.

To guide the analysis, the following hypotheses were formulated:

**Null hypothesis** **(H_0_).**
*There are no statistically significant differences in SCHO-related knowledge or food label interpretation among participants with different educational levels or dietary patterns.*


**Alternative hypothesis** **(H_1_).**
*Statistically significant differences exist in SCHO-related knowledge and food label interpretation based on educational level and dietary behavior.*


This study focuses on young adults aged 18–30 years, a life stage during which lifestyle behaviors—including dietary habits and oral health-related practices—are consolidated. This age group is particularly exposed to increased consumption of sugar-rich foods and beverages, while simultaneously demonstrating a strong capacity to integrate nutritional education and preventive interventions [5,6,7,8,9,10,11,12,13,14,15,16,17].

In Romania, demographic data from the National Institute of Statistics indicate that the 18–24 age group is numerically larger than the 25–29 age group [18]. Moreover, previous public health surveys have reported a higher participation of female respondents, a pattern commonly observed in nutritional and health-related research [17,19,20].

By examining this population, the present study aims to identify gaps in awareness and behavior related to simple carbohydrate (SCHO) intake and food label interpretation. The findings are intended to support the development of personalized preventive strategies in dentistry, with potential applications in general oral health and orthodontic care, and to inform broader public health initiatives targeting nutritional education to reduce metabolic and oral disease burden among young adults.

## 2. Materials and Methods

### 2.1. Study Design and Setting

This cross-sectional observational study was conducted between May and June 2023 to assess the level of knowledge, attitudes, and behaviors related to the consumption and labeling of simple carbohydrates (SCHO) among young adults. The study was conducted with the approval of the Ethics Committee, according to Approval No. 300/04.05.2023, issued by the “Grigore T. Popa” University of Medicine and Pharmacy in Iași. A preliminary semi-structured questionnaire, incorporating multiple-choice and Likert-scale items, was developed based on relevant scientific literature to evaluate dietary habits, SCHO intake patterns, and food label-reading behaviors. The survey was administered online through the Google Docs platform.

Informed consent was obtained from all participants prior to enrollment. Respondents were fully informed about the purpose of this study, the voluntary nature of participation, and the intended use of their data. Participation was anonymous, no identifiable personal information was collected, and confidentiality was ensured throughout this study.

### 2.2. Study Sample

According to the most recent census data (INSSE, 2021), Iași Municipality has an estimated population of approximately 300,000 inhabitants, of which roughly 10% are young adults aged 18–30 years. This corresponds to an estimated target population of approximately 30,000 individuals [18]. The minimum required sample size was calculated using the formula *N* = (z^2^ × p × (1 − p))/e^2^, where z = 1.96 (for a 95% confidence level), e = 0.05 (margin of error), and p = 0.10 (estimated population proportion). The resulting minimum sample size was *N* = 138. After applying the finite population correction for N = 30,000, the adjusted minimum sample size remained approximately 138 participants [21].

A final sample of 150 young adults was included in this study, exceeding the minimum statistical requirement. Inclusion criteria consisted of being aged 18–30 years and providing informed consent to participate. Exclusion criteria included ages outside the defined interval or refusal to provide consent. All questionnaires were completed online.

### 2.3. Study Instrument Development

The questionnaire assessed participants’ knowledge, attitudes, and behaviors related to the interpretation of nutritional information on food labels, with a specific emphasis on SCHO content. The instrument was developed through an extensive review of the scientific literature and adapted to the national context. Content validity was evaluated by a panel of academic experts from the Faculty of Dental Medicine. The questionnaire was then pretested on a pilot group of 20 individuals to assess clarity, comprehensibility, and completion time.

To evaluate the psychometric properties of the instrument, the questionnaire underwent content assessment, reliability testing, and internal consistency analysis. Exploratory factor analysis was conducted using the Kaiser–Meyer–Olkin (KMO) measure of sampling adequacy (KMO = 0.617) and Bartlett’s test of sphericity (*χ*^2^ = 429.086, df = 120, *p* = 0.000), confirming that the data were suitable for factor analysis. Internal consistency was assessed using Cronbach’s alpha. The coefficient obtained for the 16 items was α = 0.432, which reflects the multidimensional structure of the instrument, incorporating items related to knowledge, attitudes, and behavioral components.

### 2.4. Questionnaire Contents

The questionnaire consisted of 20 items structured into four sections. The first section included four socio-demographic questions assessing age, gender, place of residence (urban/rural), and highest level of education completed.

The remaining items were conceptually organized into three main domains—knowledge, attitudes, and behaviors—related to simple carbohydrate (SCHO) consumption and food label interpretation.

For analytical purposes, questionnaire items were grouped into three conceptual domains: knowledge (Q11, Q16, Q17), attitudes (Q5, Q13, Q14, Q18, Q20), and behaviors (Q6, Q7, Q8, Q9, Q10, Q12, Q15, Q19).

Knowledge domain items assessed participants’ awareness and understanding of SCHO and nutrition-related information, including awareness of SCHO classification (Q11), awareness of the harmful effects of SCHO on oral health (Q16), and recognition of the relationship between excessive SCHO consumption and dental caries (Q17).

Attitude domain items evaluated participants’ perceptions and beliefs regarding SCHO consumption and food labeling, including perceived importance of SCHO consumption in the daily diet (Q5), agreement with the statement “Food labeling is truthful” (Q13), perceived importance of being aware of sugar content in foods (Q14), and agreement with the statement “Taste is one of the main factors determining the purchase of a food product” (Q20).

Behavior domain items assessed self-reported dietary and health-related practices, including frequency of reading nutritional values on food labels (Q6), frequency of consuming sweets and carbonated beverages (Q12), frequency of dental visits (Q15), willingness to reduce SCHO intake to improve oral health (Q18), as well as behavioral choices related to SCHO consumption and food purchasing (Q7, Q8, Q9, Q10, Q19).

Response formats included Likert-type scales, frequency scales, dichotomous responses, and multiple-choice options, depending on the construct assessed. The full questionnaire is provided as Appendix A.

### 2.5. Statistical Analysis

All data were entered into a database and analyzed using IBM SPSS Statistics for Windows, Version 26.0 (IBM Corp., Armonk, NY, USA). Descriptive statistics, including frequency distributions, percentages, and mean values, were used to characterize the study population.

To examine relationships between variables and enhance the analytical rigor of this study, inferential statistical analyzes were conducted. Cross-tabulations were applied to explore associations between categorical variables, followed by comparative analyzes using the Chi-square test. In addition, correlations between knowledge- and attitude-related variables were assessed using Pearson’s correlation coefficient in order to identify the strength and direction of associations between key constructs.

The normality of continuous variables was verified using the Shapiro–Wilk test, which indicated a normal distribution (*p* > 0.05 for all variables), allowing the use of parametric procedures. A *p*-value ≤ 0.05 was considered statistically significant.

Internal consistency was assessed using Cronbach’s alpha for each conceptual domain of the questionnaire. The knowledge subscale showed a Cronbach’s alpha of 0.468, the attitudes subscale an alpha of 0.629, and the behaviors subscale an alpha of 0.471. These values indicate modest internal consistency, which can be attributed to the multidimensional and heterogeneous nature of the constructs assessed, as each domain comprised items addressing distinct aspects of SCHO-related knowledge, attitudes, and behaviors rather than a single latent trait.

## 3. Results

### 3.1. Demographic Data

The study sample included 150 young adults aged 18–30 years. Among them, 42.7% (*N* = 64) were aged 18–21 years, 39.3% (*N* = 59) were aged 22–25 years, and 18.0% (*N* = 27) were aged 26–30 years. Most participants were female (72.0%, *N* = 108), while males represented 28.0% (*N* = 42). Regarding place of residence, 78.7% (*N* = 118) lived in urban areas and 21.3% (*N* = 32) lived in rural areas. Educational levels varied, with 2.7% (*N* = 4) having completed primary school, 41.3% (*N* = 62) high school, 42.7% (*N* = 64) university studies, and 13.3% (*N* = 20) postgraduate programs (Table 1).

### 3.2. Educational Level

Regarding educational attainment, 42.7% (*N* = 64) of participants reported having completed university studies, 41.3% (*N* = 62) had completed high school, 13.3% (*N* = 20) held postgraduate degrees, and 2.7% (*N* = 4) had completed only primary education (Table 1).

Reliability analysis of the 16 questionnaire items yielded a Cronbach’s alpha coefficient of 0.432, indicating limited internal consistency—expected given the multidimensional structure of the exploratory instrument, which included items assessing knowledge, attitudes, and behaviors.

### 3.3. Perceived Importance of SCHO Consumption

When participants were asked about the perceived importance of simple carbohydrate (SCHO) consumption in daily dietary habits, 22.0% (*N* = 33) considered it very important, 41.3% (*N* = 62) important, 26.7% (*N* = 40) moderately important, 7.3% (*N* = 11) slightly important, and 2.7% (*N* = 4) not important at all. Inferential statistical analysis using the Chi-square test revealed a statistically significant association between perceived importance of SCHO consumption and age group (*χ*^2^ = 18.48, *p* = 0.047). In contrast, no statistically significant associations were observed with gender (*χ*^2^ = 3.82, *p* = 0.576) or residential background (*χ*^2^ = 0.98, *p* = 0.964) (Table 2).

Overall, these findings indicate that younger age groups perceive SCHO consumption as more important, while gender and residential background do not appear to influence this perception.

### 3.4. Frequency of Reading Nutritional Information on Product Labels

Regarding the frequency of reading nutritional information on food labels, only 13.3% (*N* = 20) of participants reported doing so very often, while 38.0% (*N* = 57) read them often. In contrast, 28.7% (*N* = 43) reported reading labels rarely, and 20.0% (*N* = 30) stated that they never read nutritional information.

Statistical analysis revealed no significant differences in label-reading frequency across age groups (*p* = 0.828, *χ*^2^ = 2.85), gender (*p* = 0.477, *χ*^2^ = 2.49), or residential background (*p* = 0.580, *χ*^2^ = 1.97) (Table 2).

### 3.5. Forms of SCHO Consumption

Responses to Q7 regarding the most frequently consumed forms of simple carbohydrates (SCHO) showed that the largest proportion of participants consumed sweets alone (27.3%, *N* = 41). Other commonly reported categories included dairy products (16.0%, *N* = 24), whole grains (10.0%, *N* = 15), and carbonated beverages (8.7%, *N* = 13). Several participants also reported consuming combinations of SCHO sources, including:Dairy and sweets (8.0%, *N* = 12);Dairy, sweets, and carbonated beverages (6.0%, *N* = 9);Dairy, whole grains, and sweets (6.0%, *N* = 9);Sweets and carbonated beverages (4.7%, *N* = 7);Whole grains and sweets (4.0%, *N* = 6);Dairy and whole grains (4.0%, *N* = 6);Dairy, whole grains, sweets, and carbonated beverages (4.0%, *N* = 6);Dairy and carbonated beverages (0.7%, *N* = 1);Whole grains, sweets, and carbonated beverages (0.7%, *N* = 1).

No statistically significant associations were identified with age (*p* = 0.875, *χ*^2^ = 16.36), gender (*p* = 0.372, *χ*^2^ = 12.96), or residential background (*p* = 0.456, *χ*^2^ = 11.87).

### 3.6. Products with Frequently Read Labels

In response to Q8, the products whose labels were most frequently read were sweets (40.7%, *N* = 61), followed by dairy products (19.3%, *N* = 29), carbonated beverages (9.3%, *N* = 14), and whole grains (6.7%, *N* = 10). Participants also reported reading labels on combinations of products, including:Whole grains and sweets (5.3%, *N* = 8);Dairy and sweets (4.7%, *N* = 7);Sweets and carbonated beverages (3.3%, *N* = 5);Whole grains, sweets, and carbonated beverages (2.7%, *N* = 4);Dairy, whole grains, sweets, and carbonated beverages (2.7%, *N* = 4);Dairy and whole grains (2.0%, *N* = 3);Dairy and whole grains (1.3%, *N* = 2);Dairy, sweets, and carbonated beverages (1.3%, *N* = 2);Whole grains and carbonated beverages (0.7%, *N* = 1).

No statistically significant differences were observed based on age (*p* = 0.701, *χ*^2^ = 19.93), gender (*p* = 0.915, *χ*^2^ = 6.02), or residential background (*p* = 0.731, *χ*^2^ = 8.67).

### 3.7. Information Most Frequently Read on Product Labels

Responses to Q9 indicated that 30.7% (*N* = 46) of participants primarily checked the expiration date on SCHO product labels. Other reported reading patterns included:Expiration date and ingredient list (12.7%, *N* = 19);Nutritional information only (11.3%, *N* = 17);Ingredient list only (10.0%, *N* = 15);Ingredient list and nutritional information (7.3%, *N* = 11);Allergen statement (3.3%, *N* = 5);Expiration date and allergen information (2.7%, *N* = 4);Expiration date and nutritional information (2.7%, *N* = 4);Expiration date, ingredients, nutritional information, and serving size (1.3%, *N* = 2);All information categories combined (1.3%, *N* = 2).

All remaining responses were below 1%.

No significant associations were found by age (*p* = 0.842, *χ*^2^ = 27.56), gender (*p* = 0.463, *χ*^2^ = 17.89), or residential background (*p* = 0.258, *χ*^2^ = 21.45).

### 3.8. Sources of Information About SCHO Consumption

Regarding Q10, the internet was the primary source of information for most participants (58.7%, *N* = 88). Other reported sources included:No information source (13.3%, *N* = 20);Internet and television (10.0%, *N* = 15);Television alone (9.3%, *N* = 14);Books and internet (6.7%, *N* = 10);Books, magazines, internet, and television (1.3%, *N* = 2);Books, magazines, and internet (0.7%, *N* = 1).

No statistically significant differences were recorded by age (*p* = 0.159, *χ*^2^ = 16.77), gender (*p* = 0.395, *χ*^2^ = 6.26), or background (*p* = 0.671, *χ*^2^ = 4.04).

### 3.9. Knowledge About the Classification of SCHO

Responses to Q11 showed varied levels of self-reported knowledge:Very informed: 10.7% (*N* = 16);Informed: 34.0% (*N* = 51);Moderately informed: 31.3% (*N* = 47);Slightly informed: 18.0% (*N* = 27);Not informed at all: 6.0% (*N* = 9).

No significant associations were identified with age (*p* = 0.304, *χ*^2^ = 9.47), gender (*p* = 0.680, *χ*^2^ = 2.31), or residential background (*p* = 0.753, *χ*^2^ = 1.91).

### 3.10. Frequency of Sweets and SCHO Beverage Consumption

Responses to Q12 showed that 18.7% (*N* = 28) of participants consumed sweets and SCHO-rich beverages very frequently, 44.0% (*N* = 66) consumed them often, 36.7% (*N* = 55) consumed them rarely, and 0.7% (*N* = 1) stated they never consumed such products.

No statistically significant differences were found by age (*p* = 0.874, *χ*^2^ = 2.45), gender (*p* = 0.528, *χ*^2^ = 2.22), or background (*p* = 0.387, *χ*^2^ = 3.03) (Table 3).

### 3.11. Trust in Food Labeling

For Q13, which assessed participants’ agreement with the statement “Food labeling is accurate,” 7.3% (*N* = 11) of respondents totally agreed, 28.0% (*N* = 42) agreed, 48.0% (*N* = 72) expressed a neutral position, 14.0% (*N* = 21) disagreed, and 2.7% (*N* = 4) totally disagreed.

Inferential statistical analysis using the Chi-square test revealed a statistically significant association between trust in food labeling and residential background (*χ*^2^ = 10.28, *p* = 0.036). In contrast, no statistically significant associations were observed with age (*χ*^2^ = 7.31, *p* = 0.503) or gender (*χ*^2^ = 1.38, *p* = 0.847) (Table 3).

These results suggest that trust in food labeling varies according to residential background, whereas age and gender do not significantly affect this perception.

### 3.12. Importance of Being Aware of Sugar Amount in Food

Most participants (60.7%, *N* = 91) considered it very important to be aware of the sugar content in food. Additionally, 22.0% (*N* = 33) rated this as important, 14.0% (*N* = 21) moderately important, 2.0% (*N* = 3) slightly important, and 1.3% (*N* = 2) not important at all.

Significant differences were found by gender (*p* = 0.005, *χ*^2^ = 15.51) and age (*p* = 0.050, *χ*^2^ = 14.88), but not by residential background (*p* = 0.859, *χ*^2^ = 1.31) (Table 3).

### 3.13. Awareness of the Link Between SCHO Consumption and Dental Caries

Regarding Q17, 77.3% (*N* = 116) of respondents reported awareness of the association between excessive SCHO consumption and dental caries, while 22.7% (*N* = 34) were unaware of this relationship.

No significant differences were identified by age (*p* = 0.090, *χ*^2^ = 4.94), gender (*p* = 0.520, *χ*^2^ = 0.41), or residential background (*p* = 0.722, *χ*^2^ = 0.13).

### 3.14. Willingness to Reduce SCHO Consumption to Improve Oral Health

Responses to Q18 showed that 18.0% (*N* = 27) were extremely likely to reduce SCHO intake to improve oral health, 48.7% (*N* = 73) likely, 21.3% (*N* = 32) neutral, 8.7% (*N* = 13) unlikely, and 3.3% (*N* = 5) extremely unlikely.

No significant differences were observed by age (*p* = 0.497, *χ*^2^ = 7.37), gender (*p* = 0.516, *χ*^2^ = 3.26), or residential background (*p* = 0.734, *χ*^2^ = 2.01).

### 3.15. Factors Influencing the Choice of SCHO Products

In Q19, taste was identified as the main factor influencing product selection (68.7%, *N* = 103), followed by nutritional value (16.7%, *N* = 25), price (12.0%, *N* = 18), and brand (2.7%, *N* = 4).

No statistically significant differences were found by age (*p* = 0.224, *χ*^2^ = 8.20), gender (*p* = 0.418, *χ*^2^ = 2.83), or background (*p* = 0.995, *χ*^2^ = 0.07).

### 3.16. The Role of Taste in Food Purchase Decisions

For Q20, 32.7% (*N* = 49) totally agreed that taste is a major factor in food purchasing decisions, 42.0% (*N* = 63) agreed, 22.0% (*N* = 33) were neutral, 2.7% (*N* = 4) disagreed, and 0.7% (*N* = 1) totally disagreed.

No significant associations were found with age (*p* = 0.285, *χ*^2^ = 9.72), gender (*p* = 0.125, *χ*^2^ = 7.21), or background (*p* = 0.106, *χ*^2^ = 7.63).

### 3.17. Correlation Between Perceived Importance of SCHO Consumption and Label Reading Frequency

The relationship between the perceived importance of simple carbohydrate (SCHO) consumption (Q5) and the frequency of reading nutritional labels (Q6) was examined using both Chi-square analysis and Pearson correlation. Among participants who considered SCHO consumption very important (*N* = 33), 8 reported reading labels very often and 13 often, whereas lower frequencies of label reading were observed among those who assigned lower importance to SCHO consumption.

Inferential statistical analysis revealed that the Chi-square test approached statistical significance (*χ*^2^ = 20.90, *p* = 0.052). In addition, Pearson’s correlation analysis indicated a weak positive association between perceived importance of SCHO consumption and frequency of label reading (r = 0.225), suggesting that individuals who attribute greater importance to SCHO consumption tend to read nutritional labels more frequently (Table 4).

Taken together, these findings suggest that greater awareness of SCHO importance is associated with more frequent engagement in label-reading behaviors.

### 3.18. Correlation Between Information Sources and Self-Perceived Knowledge of SCHO Classification

The relationship between sources of information regarding simple carbohydrate (SCHO) consumption (Q10) and self-perceived knowledge of SCHO classification into beneficial and harmful types (Q11) was examined using inferential statistical analyses. Across all levels of perceived knowledge, the internet emerged as the predominant information source. Among participants who considered themselves very informed (*N* = 16), most reported the internet as their primary source (*N* = 11). Similarly, among those who reported being informed (*N* = 51), 33 primarily relied on online sources.

Chi-square analysis revealed a statistically significant association between information source and self-perceived knowledge level (*χ*^2^ = 47.14, *p* = 0.003). However, Pearson correlation analysis indicated a very weak inverse relationship between the two variables (r = −0.063). This finding suggests that, despite statistical significance, the strength of the association is minimal and may reflect large frequency differences across categories rather than a meaningful effect size (Table 5).

The results highlight the dominant role of online sources in shaping self-perceived nutritional knowledge, despite the weak strength of the observed association.

## 4. Discussion

Overall, the findings highlight a pattern commonly reported in public health and nutritional literacy research, namely that knowledge alone does not necessarily translate into healthier dietary behaviors. Despite acknowledging the importance of monitoring sugar intake and understanding the association between SCHO consumption and dental caries, many participants reported limited engagement in food label reading and continued to prioritize taste over nutritional value when making food choices.

In this context, front-of-package labeling systems, particularly when supported by targeted nutritional literacy initiatives, may represent an effective policy-oriented tool for guiding healthier food choices among young adults. University settings offer a strategic environment for integrating nutrition education and preventive interventions, as students are in a formative life stage during which long-term dietary habits are established. Strengthening nutrition education within higher education curricula may therefore enhance the effectiveness of labeling policies and support healthier decision-making at both individual and population levels.

The results further suggest that demographic variables—particularly educational level, gender, and age—may influence certain aspects of nutritional awareness, including perceived importance of sugar monitoring and trust in food labeling. However, the behavioral indicators, such as frequency of reading labels or type of products consumed, showed minimal demographic variation. This disconnect underlines the complexity of behavioral change in young adults, a population characterized by autonomy in food choices, strong marketing exposure, and variable nutritional knowledge.

In the following subsections, each major domain of the questionnaire is discussed in relation to existing literature, emphasizing its relevance for both general health and preventive oral health strategies.

### 4.1. Perceived Importance of SCHO in the Diet (Q5)

The present study found that only 21.3% of participants considered the consumption of simple carbohydrates (SCHO) to be very important in their daily diet. This relatively low perception of importance is concerning, given the World Health Organization (WHO) recommendation to limit free sugar intake to less than 10%—and ideally below 5%—of total daily caloric intake to reduce the risk of metabolic and oral diseases [5].

The discrepancy between awareness of sugar-related health risks and the perceived importance of SCHO intake may reflect insufficient nutritional education during adolescence and young adulthood. Although many participants recognized the cariogenic potential of sugars, their broader systemic consequences—including obesity, insulin resistance, and non-alcoholic fatty liver disease—appear to be underestimated in this age group.

From an orthodontic perspective, this gap in perception is clinically relevant. High SCHO intake in patients wearing fixed appliances significantly increases the risk of enamel demineralization and white spot lesions due to plaque accumulation and reduced self-cleaning capacity. Personalized preventive strategies in orthodontics rely heavily on accurate assessment of dietary risk, and low perceived importance of sugar intake may undermine compliance.

These findings are consistent with previous literature indicating a disconnect between knowledge and attitudes related to sugar consumption [21,22]. A recent Romanian study among adolescents reported that frequent intake of sugary snacks was strongly associated with higher caries prevalence [23]. In orthodontic patients, such patterns carry even greater risk due to the combined effects of dietary behavior, microbial biofilm retention, and mechanical difficulties related to appliance maintenance.

### 4.2. Frequency and Content of Label Reading (Q6, Q8, Q9)

Only 13.3% of respondents reported reading nutrition labels very often, and most participants focused primarily on expiration dates (30.7%), with far fewer examining nutritional information (11.3%) or ingredient lists (10%). These percentages are noticeably lower than those reported by Cooper [24], who found that more than 75% of consumers routinely check ingredient lists, and by Saidakowska, who observed that sugar content (75.4%) and caloric values (65.2%) were among the most frequently consulted elements [6].

Several factors may explain the limited engagement with nutritional labeling in the current study. The complexity and lack of clarity in labeling formats—particularly the absence of indicators such as glycemic index, glycemic load, or differentiation between natural and added sugars—reduces the usefulness of labels for the general public. Additionally, ambiguous marketing claims such as “natural sugar,” “no added sugar,” or “light” may mislead consumers and contribute to misunderstandings, especially among individuals with low nutritional literacy.

For orthodontic patients, limited use of nutrition labels has direct clinical implications. Fixed appliances create retention areas that favor bacterial accumulation, making sugar control a critical aspect of caries prevention during treatment. The inability to correctly interpret labels may lead to unintentional overconsumption of free sugars, thereby increasing the risk of enamel demineralization and white spot lesions.

Personalized orthodontic care should therefore incorporate structured nutritional counseling focused on practical label interpretation. Integrating visual aids, simplified label-reading guides, and digital applications tailored to individual literacy levels may significantly improve patients’ ability to identify high-sugar products and adhere to preventive recommendations.

### 4.3. Knowledge About Sugar Classification and Label Trust (Q11, Q13, Q14)

Less than half of the participants in this study reported being informed about the classification of sugars into beneficial and harmful types. At the same time, although 60.7% considered it very important to be aware of sugar content, 48.0% expressed uncertainty regarding the accuracy of food labeling. This lack of trust may stem from the absence of standardized sugar-related disclosures on packaging and the influence of contradictory or misleading marketing claims.

Evidence from Poland suggests that young consumers increasingly demand transparency in labeling—particularly regarding sugar content—due to heightened concerns over obesity and other noncommunicable diseases [6]. Improved legislative clarity, especially the distinction between total sugars and added sugars, could enhance consumer confidence and support more informed purchasing decisions.

From an orthodontic standpoint, this issue is particularly significant. Patients undergoing treatment with fixed appliances face an elevated risk of enamel demineralization, and the inability to distinguish between harmful added sugars and naturally occurring sugars complicates preventive dietary guidance. Personalized patient education that focuses on identifying cariogenic products—moving beyond ambiguous industry terms such as “no added sugar”—can empower individuals to make informed, healthier choices. Implementing individualized educational strategies may improve compliance, lower caries risk, and ultimately support more favorable orthodontic treatment outcomes.

### 4.4. Frequency and Patterns of SCHO Consumption (Q7, Q12, Q17)

Participants reported frequent consumption of sweets, dairy products, and carbonated beverages. In total, 18.7% stated that they consumed such items very often, and 44.0% reported consuming them often, while only 0.7% indicated that they never consumed SCHO-rich products. Although 77.3% of respondents acknowledged the link between sugar intake and dental caries, this awareness did not translate into lower consumption levels, highlighting the persistent challenge of converting knowledge into behavior.

Existing literature consistently demonstrates a strong association between high sugar intake and the development of dental caries [25]. Romanian studies echo these findings; one investigation among adolescents reported that 42.1% consumed sugary foods more than once per day [26]. This alignment with national data suggests that frequent sugar consumption remains a widespread behavioral pattern across age groups.

The harmful effects of sugar-rich foods—including carbonated beverages, fruit juices, and sweet snacks—are well documented. These products contribute to pronounced drops in dental plaque pH, increasing the risk of enamel demineralization and caries formation [25,27,28]. Consistent with previous research, our study found that 18.7% of participants consumed sweets and soft drinks very often, 44.0% consumed them often, and only a negligible proportion (0.7%) reported complete avoidance.

These findings emphasize the need for targeted educational interventions that go beyond raising awareness. Behavioral strategies, motivational interviewing, and personalized dietary counseling may be required to effectively reduce SCHO intake—particularly among young adults who exhibit high-risk consumption patterns.

### 4.5. Motivation to Reduce SCHO Intake and Role of Taste (Q18, Q19, Q20)

Although most participants acknowledged the health risks associated with excessive sugar intake, only 18% reported being highly motivated to reduce their consumption. Taste emerged as the dominant factor influencing the choice of SCHO-rich products (68.7%), whereas nutritional value (16.7%) and price (12.0%) played considerably smaller roles.

These findings are consistent with previous research demonstrating that taste is one of the strongest predictors of food choice, even when individuals are aware of associated health concerns [29,30,31,32]. This suggests that simply providing nutritional information may be insufficient to drive behavioral change. Effective preventive strategies must also address environmental and psychological determinants of food choice, ensuring that healthier options are both appealing and accessible.

In orthodontic patients, this behavioral trend has direct clinical relevance. Frequent sugar consumption in the presence of fixed appliances substantially increases the risk of enamel demineralization and the development of white spot lesions. Personalized interventions that align dietary recommendations with patients’ preferences are therefore essential. Potential strategies include the reformulation of commonly consumed products to reduce sugar content without compromising taste, as well as the implementation of intuitive visual aids—such as color-coded front-of-pack labeling—to support healthier choices.

Within orthodontic practice, these measures should be complemented by individualized counseling that links dietary behaviors to treatment progress and long-term oral health outcomes. Strengthening patient motivation by emphasizing the visible and tangible benefits of reduced SCHO intake may enhance adherence and improve overall treatment success.

### 4.6. Diet Type and Oral Health (Q16)

Emerging evidence indicates that overall dietary patterns influence oral health beyond the effects of sugar intake alone. While vegetarian diets are associated with systemic health benefits, they may also increase the risk of dental erosion due to the frequent consumption of acidic foods and beverages [33]. Conversely, protective dietary models—such as anti-inflammatory or high-fiber diets—have been shown to support oral health by modulating the oral microbiome, enhancing salivary function, and reducing inflammatory responses [34,35].

These considerations are particularly relevant for orthodontic patients. The combined effects of dietary habits, salivary pH fluctuations, and biofilm retention around brackets and wires can substantially modify the risk of caries and erosion. For this reason, integrating a personalized dietary assessment into orthodontic care is essential. Such an approach should evaluate not only the quantity and frequency of SCHO consumption but also broader dietary characteristics, including acidity, fiber intake, and the presence of protective nutrients.

Tailored dietary counseling enables clinicians to identify individual nutritional risk factors and guide patients towards choices that promote both systemic and oral health. This strategy supports the broader concept of personalized preventive care in orthodontics, where individualized dietary recommendations complement mechanical plaque control and fluoride-based interventions. By addressing diet holistically, clinicians can better optimize treatment outcomes and reduce the likelihood of long-term complications such as caries, erosion, and white spot lesion development.

### 4.7. Dental Visit Frequency and Behavioral Barriers (Q15)

In this study, only 36% of participants reported attending biannual dental check-ups, while 14% sought dental care only in emergencies and 1.3% reported never visiting the dentist. These findings indicate suboptimal preventive behaviors among young adults—an issue of particular concern for orthodontic patients, who require regular monitoring to minimize treatment-related complications such as enamel demineralization, gingival inflammation, and progression of white spot lesions.

Existing literature demonstrates that psychological factors—including fear of judgment, loss of control, anticipated pain, and low trust in dental professionals—often pose stronger barriers to dental attendance than financial or logistical constraints [36,37,38,39]. Despite this evidence, most interventions continue to focus primarily on reducing treatment costs or improving access, while overlooking the emotional and behavioral determinants of dental avoidance [40,41,42].

Within orthodontic care, addressing these barriers is essential to ensure treatment success. Missed appointments can impede appliance adjustments, extend treatment duration, and heighten the risk of caries or white spot lesions due to insufficient clinical supervision. Personalized recall systems, motivational interviewing, and patient-centered communication strategies that acknowledge individual fears, expectations, and preferences may significantly enhance compliance. By integrating psychological considerations into preventive and therapeutic planning, orthodontists can improve adherence, optimize treatment outcomes, and support sustained oral health beyond the completion of active therapy.

### 4.8. Sources of Nutritional Information and Perceived Knowledge (Q10, Q11 Correlation)

The internet was the most frequently cited source of nutrition information (58.7%). However, a statistically significant negative correlation was found between the number of information sources and participants’ perceived knowledge of sugar classification (r = −0.063, *p* = 0.003). This finding suggests that access to multiple sources does not necessarily enhance understanding. Instead, it may contribute to information overload or increase exposure to contradictory or misleading content, particularly in digital environments where the quality of health information varies widely. These results highlight the need to incorporate digital literacy into nutritional education initiatives, enabling young adults to critically evaluate online content and distinguish evidence-based information from unreliable sources.

### 4.9. Study Limitations

Although the sample size meets the minimum statistical requirements, it remains relatively small and non-randomized, which limits the external validity of the findings. The marked overrepresentation of women (72%) and the underrepresentation of individuals aged 26–30 years introduce demographic imbalances that may have influenced the distribution of behaviors and knowledge patterns. Furthermore, the online self-administration of the questionnaire may have selectively favored participants with higher digital literacy and greater health awareness, resulting in potential selection bias.

Although the sample size exceeded the minimum statistical requirement, the final number of participants remains relatively small. This limitation may affect the generalizability of the findings and reduce the statistical power to detect smaller effect sizes. Additionally, the relatively small sample size limited the feasibility of conducting multivariate analyzes to identify independent predictors of label-reading behaviors and simple carbohydrate consumption patterns.

An important methodological limitation of the present study concerns the relatively low internal consistency of the questionnaire, as reflected by the Cronbach’s alpha coefficient (α = 0.432), particularly within the knowledge- and behavior-related subscales. This finding can be partially attributed to the multidimensional nature of the instrument, which was designed to capture various constructs—including knowledge, attitudes, and behaviors—related to simple carbohydrate consumption, food label interpretation, and oral health-related practices.

In such exploratory and conceptually broad instruments, Cronbach’s alpha may underestimate internal consistency, as it assumes unidimensionality and equal contribution of all items—conditions that are not fully met in questionnaires assessing complex behavioral patterns. Therefore, the observed alpha values should not be interpreted as indicating poor measurement quality, but rather as reflecting the conceptual heterogeneity of the assessed domains. Nevertheless, this limitation should be considered when interpreting the findings. Future research should focus on refining and validating more homogeneous, domain-specific scales, ideally using larger samples and confirmatory factor analysis to improve reliability and measurement precision.

Additionally, low expected frequencies observed in several statistical tests, along with borderline internal consistency in certain sections, may have reduced the robustness of some associations. These constraints highlight the need for caution when generalizing findings to broader populations.

Future studies should therefore include larger and more diverse samples, employ probability sampling strategies, and integrate objective clinical indicators to enhance the accuracy and applicability of conclusions. Such methodological improvements would strengthen the evidence base required to support personalized preventive strategies in orthodontics and general oral health.

Additionally, the online and self-reported nature of the survey may have introduced recall bias and social desirability bias, potentially influencing participants’ responses. Moreover, due to the cross-sectional design of this study, causal relationships between dietary behaviors, nutritional literacy, and oral health outcomes cannot be established. These limitations should be considered when interpreting the findings; however, they do not affect the overall conclusions of this study.

### 4.10. Implications and Recommendations

Despite recognizing the health consequences associated with excessive simple carbohydrate (SCHO) intake, most participants continued to prioritize taste in their food choices and demonstrated limited engagement in preventive oral health behaviors. This discrepancy between knowledge and behavior underscores the need for multifaceted public health interventions that integrate nutritional education with effective behavior change strategies.

Improving the clarity and transparency of food labeling—particularly regarding added sugars—could support healthier consumer choices. Strengthening regulations governing sugar-related claims and promoting standardized labeling formats may also reduce consumer confusion and enhance trust in nutritional information. Additionally, integrating oral health considerations into broader nutrition policies, especially those targeting young adults, may yield synergistic benefits for both systemic and dental health.

From a preventive dentistry and orthodontic perspective, personalized counseling is essential. Tailoring dietary guidance to individual preferences, literacy levels, and motivational profiles can foster better adherence to recommendations, especially among patients at higher risk of enamel demineralization.

### 4.11. Future Research Directions

Future research should build upon the present findings by including larger and more socio-demographically diverse samples and by employing probability-based sampling methods to enhance generalizability. Importantly, integrating objective clinical indicators alongside self-reported dietary behaviors and nutritional literacy measures would strengthen the validity of future investigations. Longitudinal studies combining questionnaire-based assessments with clinical oral health outcomes—such as dental caries prevalence, the DMFT index, plaque index, or salivary parameters—could provide a more comprehensive understanding of the relationships between simple carbohydrate consumption, food label-reading behaviors, and oral health status. Incorporating such clinical measures would allow for the validation of self-reported data, facilitate the identification of clinically meaningful risk profiles, and support the development of evidence-based preventive strategies in dentistry and interdisciplinary public health interventions.

## 5. Conclusions

This study highlights a persistent discrepancy between young adults’ awareness of the health risks associated with excessive simple carbohydrate (SCHO) consumption and their actual dietary and preventive behaviors. Although most participants recognized the harmful effects of high sugar intake—including its association with dental caries—food choices continued to be driven predominantly by taste preferences, with limited engagement in nutrition label reading.

These findings underscore the need to strengthen nutritional education and enhance the clarity, accuracy, and accessibility of food labeling to support healthier decision-making. Integrating oral health considerations into broader nutrition and public health initiatives may further contribute to reducing sugar-related systemic and dental diseases among young adults.

Future research should involve larger, more demographically balanced populations and incorporate objective measures—such as standardized dental clinical indices—to provide a more comprehensive understanding of the relationship between dietary habits, nutritional literacy, and oral health outcomes. Such evidence will be essential for developing tailored, effective preventive strategies that promote long-term health behaviors in this age group.

## Figures and Tables

**Table 1 nutrients-18-00091-t001:** Sociodemographic characteristics of the study population.

Variable	Category	*N* (%)
Sex	Male	42 (28.0)
	Female	108 (72.0)
Age	18–21	27 (18.0)
	22–25	64 (42.7)
	26–30	59 (39.3)
Residence	Urban	118 (78.7)
	Rural	32 (21.3)
Education	Primary school	4 (2.7)
	High school	62 (41.3)
	University	64 (42.7)
	Postgraduate	20 (13.3)

Note: Values represent frequencies and percentages relative to the total sample (N = 150). Statistical comparisons between demographic variables and SCHO-related outcomes are presented in subsequent sections.

**Table 2 nutrients-18-00091-t002:** Distribution of responses to Q5–Q6–Q11 regarding importance of SCHOOL, food label reading, and knowledge on SCHO classification.

No	Question	*N* (%)	A		G		B	
			*χ* ^2^	*p*	*χ* ^2^	*p*	*χ* ^2^	*p*
Q5	Not important at all	4 (2.67)	18.48	0.047 *	3.82	0.576	0.98	0.964
	Less important	11 (7.33)						
	Moderately important	40 (26.67)						
	Important	62 (41.33)						
	Very important	33 (22)						
Q6	Never	30 (20)	2.85	0.83	2.49	0.48	1.97	0.58
	Quite Rarely	28.67 (43)						
	Often	38 (57)						
	Very often	13.33 (20)						
Q11	Not informed at all	9 (6.0)	9.47	0.304	2.31	0.68	1.91	0.753
	Less informed	27 (18.0)						
	Moderately informed	47 (31.3)						
	Informed	51 (34.0)						
	Very informed	16 (10.7)						

Notes: A = age; G = gender; B = background; *χ*^2^ = Pearson Chi-square; * *p* ≤ 0.05 is considered statistically significant. Q5 = Perceived importance of SCHO in daily diet; Q6 = Frequency of reading nutritional values on labels; Q11 = Knowledge of SCHO classification.

**Table 3 nutrients-18-00091-t003:** Distribution of responses to Q12–Q16 regarding consumption of sweets and SCHO drinks, perceptions of food labeling, sugar awareness, and knowledge of SCHO effects on the oral environment.

No	Question	*N* (%)	A		G		B	
			*χ* ^2^	*p*	*χ* ^2^	*p*	*χ* ^2^	*p*
Q12	Never	1 (0.67)	2.45	0.874	2.22	0.528	3.03	0.387
	Quite rarely	55 (36.67)						
	Often	66 (44.00)						
	Very often	28 (18.67)						
Q13	Totally agree	11 (7.33)	7.31	0.503	1.38	0.847	10.28	0.036 *
	Agree	42 (28.00)						
	Neither disagree nor agree	72 (48.00)						
	Disagree	21 (14.00)						
	Totally disagree	4 (2.67)						
Q14	Not important at all	2 (1.33)	15.51	0.050 *	14.88	0.005 *	1.31	0.859
	Less important	3 (2.00)						
	Moderately important	21 (14.00)						
	Important	33 (22.00)						
	Very important	91 (60.67)						
Q16	Not important at all	0 (0.00)	8.03	0.24	3.3	0.35	8.81	0.42
	Less important	15 (10.00)						
	Moderately important	43 (28.67)						
	Important	57 (38.00)						
	Very important	35 (23.33)						

N = count; A = age; G = gender; B = background; *χ*^2^ = Pearson Chi-Square; *p* = significance; * *p* ≤ 0.05 is considered statistically significant; Q12 = How often do you consume sweets and SCHO drinks? Q13 = Do you agree with the statement ‘Food labeling is truthful’? Q14 = How important do you think it is to be aware of the amount of sugar in food? Q16 = How informed are you about the harmful effects of SCHO on the oral environment?

**Table 4 nutrients-18-00091-t004:** Correlation between responses to Q5 and Q6 regarding the importance of SCHO consumption and the frequency of reading nutritional values on product labels.

		Q6				Total	*χ* ^2^	*p*	r
		Very Often	Often	Rarely	Never				
Q5	Very important	8	13	5	7	33	20.90	0.052	0.225
	Important	7	28	19	8	62			
	Moderately important	4	13	13	10	40			
	Slightly important	1	2	6	2	11			
	Not important at all	0	1	0	3	4			

*χ*^2^ = Pearson Chi-Square; *p* = Asymptotic Significance (2-sided); r = Pearson correlation coefficient; Q5 = How important do you think the consumption of SCHO is in the daily diet? Q6 = How often do you usually read the nutritional values on product labels?

**Table 5 nutrients-18-00091-t005:** Correlation between Q10 (“From which sources do you gather information about SCHO consumption?”) and Q11 (“How informed are you about the classification of SCHO into beneficial and harmful?”).

		Q11					Total	*χ* ^2^	*p*	r
		Very	Informed	Moderately	Slightly	Not				
Q10	Internet	11	33	24	16	4	88	47.14	0.003	−0.063
	TV	0	2	8	2	2	14			
	I don’t gather information	0	6	3	8	3	20			
	Books, Internet	2	4	4	0	0	10			
	Internet, TV	1	6	7	1	0	15			
	Books, Magazines, Internet	0	0	1	0	0	1			
	Books, Magazines, Internet, TV	2	0	0	0	0	2			

*χ*^2^ = Pearson Chi-Square; *p* = Asymptotic Significance (2-sided); r = Pearson correlation coefficient, Q10 = From which sources do you gather information about SCHO consumption; Q11 = How informed are you that SCHO are classified as beneficial and harmful?

## Data Availability

The processed data is available at the following link: https://drive.google.com/drive/folders/1ABdPgKdUJWzKg-0Nbb26DdUfPbsruCEn?usp=drive_link (accessed on 23 December 2025).

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
