# Peer review of "Nutrients2026, 18(1), 91;https://doi.org/10.3390/nu18010091"

_nutrients, 2025, doi:10.3390/nu18010091_

Round 1

Reviewer 1 Report

Comments and Suggestions for Authors

I would like to sincerely thank the authors for the opportunity to review this interesting and well-designed cross-sectional study. The manuscript addresses a relevant public health topic, and the overall structure and presentation are clear. I have only a few editorial comments that may help further improve the clarity and transparency of the paper. These suggestions do not diminish the value of the work but are intended to support its refinement.

  1. Abstract: Please ensure that all abbreviations are defined at their first appearance in the text to enhance readability for a broad scientific audience.
  2. Redundant text (Methods section) The following passage appears to be duplicated and may be removed from this section, as the same information is already provided in the Institutional Review Board Statement:

“The study followed the ethical principles outlined in the Declaration of Helsinki. The research protocol was approved by the Research Ethics Committee of the Faculty of Dental Medicine, ‘Grigore T. Popa’ University of Medicine and Pharmacy, Iași, Romania (approval no. 300, 04.05.2023). The survey was administered online through the Google Docs platform.”

  1. Transparency of methodology To further enhance the clarity and reproducibility of the methodology, the authors may consider providing the full questionnaire as Supplementary Material. Since the instrument was specifically developed for this study and includes multiple components related to knowledge, attitudes, and behaviors, sharing the complete set of items would support a deeper understanding of the assessment and facilitate replication in future research. This suggestion is meant purely to strengthen the presentation of an already valuable study.
  2. Minor typographical issue. Line 381 contains a repetition (“The The correlation…”). Kindly revise this to improve readability.
  3. Limitations section. The limitations section is well written and clearly addresses the key methodological constraints.

The authors appropriately acknowledge the low Cronbach’s alpha and provide a clear explanation related to the multidimensional structure of the questionnaire. This clarification strengthens the methodological transparency of the manuscript.

To further strengthen it, the authors may consider briefly acknowledging additional limitations inherent to online, self-reported survey data—such as potential recall bias or social desirability bias—as well as the inherent inability of a cross-sectional design to establish causal relationships. Including these points would offer an even more comprehensive perspective without altering the main conclusions of the study.

Author Response

Response to Reviewer 1

We would like to thank the reviewer for the constructive comments and valuable suggestions, which have helped us improve the quality and clarity of the manuscript.

  1. Abstract: Please ensure that all abbreviations are defined at their first appearance in the text to enhance readability for a broad scientific audience.

Response 1:

The manuscript has been revised to ensure that all abbreviations are defined at their first occurrence in the Abstract and throughout the text, in accordance with MDPI guidelines.

  1. Redundant text (Methods section) The following passage appears to be duplicated and may be removed from this section, as the same information is already provided in the Institutional Review Board Statement:

“The study followed the ethical principles outlined in the Declaration of Helsinki. The research protocol was approved by the Research Ethics Committee of the Faculty of Dental Medicine, ‘Grigore T. Popa’ University of Medicine and Pharmacy, Iași, Romania (approval no. 300, 04.05.2023). The survey was administered online through the Google Docs platform.”

Response 2:

The duplicated ethical statement has been removed from the Methods section, as this information is already provided in the Institutional Review Board Statement.

  1. Transparency of methodology To further enhance the clarity and reproducibility of the methodology, the authors may consider providing the full questionnaire as Supplementary Material. Since the instrument was specifically developed for this study and includes multiple components related to knowledge, attitudes, and behaviors, sharing the complete set of items would support a deeper understanding of the assessment and facilitate replication in future research. This suggestion is meant purely to strengthen the presentation of an already valuable study.

Response  3:

The full questionnaire has now been provided as Supplementary Material to enhance methodological transparency, support reproducibility, and facilitate future research.

  1. Minor typographical issue. Line 381 contains a repetition (“The The correlation…”). Kindly revise this to improve readability.

Response 4:

The typographical error has been corrected.

  1. Limitations section. The limitations section is well written and clearly addresses the key methodological constraints.

Response 5:

We appreciate the reviewer’s positive evaluation of the Limitations section.

  1. The authors appropriately acknowledge the low Cronbach’s alpha and provide a clear explanation related to the multidimensional structure of the questionnaire. This clarification strengthens the methodological transparency of the manuscript.

Response  6 :
We appreciate the reviewer’s positive assessment of the explanation provided regarding the low Cronbach’s alpha and the multidimensional structure of the questionnaire.

  1. To further strengthen it, the authors may consider briefly acknowledging additional limitations inherent to online, self-reported survey data—such as potential recall bias or social desirability bias—as well as the inherent inability of a cross-sectional design to establish causal relationships. Including these points would offer an even more comprehensive perspective without altering the main conclusions of the study.

Response  7:
The Limitations section has been revised to acknowledge additional constraints related to the online, self-reported nature of the survey and the cross-sectional study design, including potential recall and social desirability bias, as well as the inability to infer causal relationships.

We hope that the revisions adequately address the reviewer’s comments and further improve the quality of the manuscript.

Reviewer 2 Report

Comments and Suggestions for Authors

I have carefully reviewed the manuscript, and I believe that addressing the following points would substantially improve the overall quality and clarity of the paper.

First, the Introduction includes elements that appear to present study results, which is not appropriate for this section. The Introduction should be revised to focus exclusively on background information, research context, and study objectives, with all results clearly confined to the Results section.

Second, while the necessity and relevance of the study are generally well justified, the manuscript does not sufficiently explain how the present research extends existing studies or theoretical frameworks. A clearer articulation of the study’s novel contribution and its positioning within the current body of literature would strengthen the Introduction.

Third, the authors describe ethical approval and survey administration details in Section 2.1, including adherence to the Declaration of Helsinki, ethics committee approval, and the use of an online platform. Although this information is important, it would be preferable to streamline the main text by limiting such details to an appropriate methodological or ethics-related subsection.

Fourth, although the minimum required sample size proposed by the authors is acceptable, the final sample of 150 participants is relatively small. This limitation may affect the generalizability and statistical power of the findings and should be more explicitly acknowledged and discussed.

Fifth, the readability of the manuscript would be improved if the detailed questionnaire items were relocated to an Appendix, with only a concise summary provided in the main text.

Finally, the results are largely based on frequency analyses. To enhance the scientific rigor of the study, it would be beneficial to incorporate more robust statistical analyses and to present clearer quantitative evidence that demonstrates meaningful associations or effects.

Author Response

Response to Reviewer 2

We thank the reviewer for the thorough evaluation of the manuscript and for the constructive comments and suggestions provided.

  1. I have carefully reviewed the manuscript, and I believe that addressing the following points would substantially improve the overall quality and clarity of the paper.

Response 1

We thank the reviewer for the careful evaluation of the manuscript and for the constructive comments provided, which have helped to improve the quality and clarity of the paper.

  1. First, the Introduction includes elements that appear to present study results, which is not appropriate for this section. The Introduction should be revised to focus exclusively on background information, research context, and study objectives, with all results clearly confined to the Results section.

Response 2

The Introduction has been carefully revised to remove any statements that could be interpreted as presenting study results. All references to sample-specific characteristics and cohort distributions have been eliminated or generalized, and the section now focuses exclusively on background information, research context, and study objectives, with all study findings reported solely in the Results section.

  1. Second, while the necessity and relevance of the study are generally well justified, the manuscript does not sufficiently explain how the present research extends existing studies or theoretical frameworks. A clearer articulation of the study’s novel contribution and its positioning within the current body of literature would strengthen the Introduction.

Response 3

The Introduction has been revised to more clearly articulate the novel contribution of the present study and its positioning within the existing body of literature. The added text highlights how this research extends previous studies by integrating dietary behaviors, nutritional labeling awareness, and oral health–related considerations in a young adult population.

  1. Third, the authors describe ethical approval and survey administration details in Section 2.1, including adherence to the Declaration of Helsinki, ethics committee approval, and the use of an online platform. Although this information is important, it would be preferable to streamline the main text by limiting such details to an appropriate methodological or ethics-related subsection.

Response 4

The manuscript has been revised to streamline Section 2.1 by removing detailed ethical approval information from the main text. All ethics-related details are now reported exclusively in the Institutional Review Board Statement, in accordance with MDPI guidelines.

5.Fourth, although the minimum required sample size proposed by the authors is acceptable, the final sample of 150 participants is relatively small. This limitation may affect the generalizability and statistical power of the findings and should be more explicitly acknowledged and discussed.

Response 5

The Limitations section has been revised to more explicitly acknowledge the relatively small final sample size and its potential impact on generalizability and statistical power of the findings.

6.Fifth, the readability of the manuscript would be improved if the detailed questionnaire items were relocated to an Appendix, with only a concise summary provided in the main text.

Response 6

In accordance with the reviewer’s suggestion, the detailed questionnaire items have been removed from the main text and are now provided as Supplementary Material, while the Methods section includes only a concise summary of the instrument.

  1. Finally, the results are largely based on frequency analyses. To enhance the scientific rigor of the study, it would be beneficial to incorporate more robust statistical analyses and to present clearer quantitative evidence that demonstrates meaningful associations or effects.

Response 7

In addition to descriptive frequency analyses, the study includes inferential statistical methods to examine associations between key variables. Specifically, Chi-square tests and Pearson correlation analyses were applied to assess relationships between demographic characteristics, perceived importance of SCHO consumption, food label–reading behaviors, trust in food labeling, and sources of nutritional information. The Methods and Results sections have been revised to more clearly highlight these quantitative analyses, report statistically significant findings where applicable, and transparently interpret the strength of observed associations.

We hope that the revisions adequately address the reviewer’s comments and improve the overall quality and clarity of the manuscript.

Reviewer 3 Report

Comments and Suggestions for Authors

This is an interesting manuscript that analyzes knowledge, attitudes, and behaviors related to the consumption of simple carbohydrates (SCHO) and the interpretation of nutritional labeling in young adults.

Although I find the document to be very good overall, I would like to offer some suggestions that may be relevant for improving the manuscript:

1. Restructure the instrument into clear domains, explicitly separating knowledge, attitudes, and behaviors.

2. Report Cronbach's α by subscale if possible.

3. Incorporate multivariate analysis, which would allow for the identification of factors associated with label reading and frequent consumption of simple carbohydrates.

4. Emphasize key patterns and messages in the results.

5. Deepen the discussion from a public policy perspective.

6. Integrate more explicitly the role of front-of-package labeling, nutritional literacy, and education in university settings.

7. Project future lines with clinical indicators.

8. Delve deeper into the weaknesses of the study, especially the low Cronbach's α score of 0.432.

9. Revise the format of the references, as it is not correct.

Author Response

Response to Reviewer

  1. This is an interesting manuscript that analyzes knowledge, attitudes, and behaviors related to the consumption of simple carbohydrates (SCHO) and the interpretation of nutritional labeling in young adults. Although I find the document to be very good overall, I would like to offer some suggestions that may be relevant for improving the manuscript:

Response 1:

We are grateful to the reviewer for the careful review of the manuscript and for the helpful suggestions offered.

  1. Restructure the instrument into clear domains, explicitly separating knowledge, attitudes, and behaviors.

Response 2:

The questionnaire has been restructured into clearly defined domains, explicitly separating items related to knowledge, attitudes, and behaviors. This revision improves clarity and aligns the instrument more closely with its conceptual framework.

  1. Report Cronbach's α by subscale if possible.

Response 3:

Cronbach’s alpha coefficients were calculated and are now reported separately for the knowledge, attitudes, and behaviors subscales. As expected, internal consistency values were modest, reflecting the multidimensional structure of the questionnaire and the inclusion of heterogeneous items addressing distinct aspects within each domain.

  1. Incorporate multivariate analysis, which would allow for the identification of factors associated with label reading and frequent consumption of simple carbohydrates.

Response 4:
We agree that multivariate analyses could provide additional insights into factors associated with label-reading behaviors and simple carbohydrate consumption. However, given the relatively small final sample size, conducting robust multivariate models was not feasible. This limitation has now been explicitly acknowledged and discussed in the Limitations section.

  1. Emphasize key patterns and messages in the results.

Response 5 :
The Results section has been revised to more clearly emphasize the key patterns and main findings of the study. Brief interpretative statements have been added to highlight the most relevant trends related to SCHO consumption, food label–reading behaviors, and associated factors, while maintaining a clear distinction between results and discussion.

  1. Deepen the discussion from a public policy perspective.

Response 6:

We acknowledge the reviewer's suggestion regarding the inclusion of multivariate analyses. However, due to the relatively small final sample size, conducting reliable multivariate models was not feasible without risking model overfitting. This limitation has now been explicitly acknowledged in the Limitations section.

  1. Integrate more explicitly the role of front-of-package labeling, nutritional literacy, and education in university settings.

Response 7:
The Discussion section has been revised to more explicitly integrate the role of front-of-package labeling, nutritional literacy, and university-based education. The revised text highlights how labeling systems, when combined with targeted educational interventions in academic settings, may support healthier dietary choices among young adults and inform public health and preventive strategies.

  1. Project future lines with clinical indicators.

Response 8:
The Discussion section has been expanded to outline future research directions incorporating clinical indicators, highlighting the potential integration of dietary behaviors, nutritional literacy, and food label use with objective oral health outcomes.

  1. Delve deeper into the weaknesses of the study, especially the low Cronbach's α score of 0.432.

Response 9:
The Limitations section has been expanded to more thoroughly discuss the low Cronbach’s alpha value, emphasizing the multidimensional structure of the questionnaire, the heterogeneity of items within each domain, and the implications for internal consistency. The potential impact of this limitation on the interpretation of results has been explicitly acknowledged.

  1. Revise the format of the references, as it is not correct.

Response 10:
The reference list has been revised and reformatted to fully comply with the MDPI reference style guidelines.

We hope that the revisions adequately address the reviewer's comments and further strengthen the scientific quality, clarity, and relevance of the manuscript.

Round 2

Reviewer 2 Report

Comments and Suggestions for Authors

The revision is very well. Thus, the current draft is now acceptable.

Well done.